

# Arctic Ocean geostrophic circulation 2003-2014

**Thomas W. K. Armitage**[1,2]**, Sheldon Bacon**[3]**, Andy L. Ridout**[1]**, Alek A. Petty**[4,5]**,
Steven Wolbach**[6]**, and Michel Tsamados**[1]

[1]Centre for Polar Obseration and Modelling, University College London, London, UK
[2]Now at Jet Propulsion Laboratory, California Institute of Technology, Pasadena, California, USA
[3]National Oceanography Centre, Southampton, UK
[4]Cryospheric Sciences Laboratory, NASA Goddard Space Flight Center, Greenbelt, Maryland, USA
[5]Earth System Science Interdisciplinary Center, University of Maryland, College Park, Maryland, USA
[6]Department of Atmopsheric and Oceanic Sciences, University of Maryland, College Park, Maryland, USA

Correspondence to: T. W. K. Armitage (Tom.W.Armitage@jpl.nasa.gov)



## Abstract

Monitoring the surface circulation of the ice-covered Arctic Ocean is generally limited in space, time or both. We present a new 12-year record of geostrophic currents at monthly resolution in the ice-covered and ice-free Arctic Ocean and characterise their seasonal to decadal variability from 2003-2014, a period of rapid environmental change in the Arctic. Geostrophic currents around the Arctic basin increased in the late '00s, with the largest increases observed in summer. Currents in the southeastern Beaufort gyre accelerated in late 2007 with higher current speeds sustained until 2011, after which they decreased to speeds representative of the period 2003-2006. The strength of the northwestward current in the southwest Beaufort gyre more than doubled between 2003 and 2014. This pattern of changing currents is linked to shifting of the gyre circulation to the northwest during the time period. The Beaufort gyre circulation and Fram Strait current are strongest in winter, modulated by the seasonal strength of the atmospheric circulation. Eddy kinetic energy is also larger in winter and we find high eddy activity congruent with features of the seafloor bathymetry. The variability of Arctic Ocean geostrophic circulation highlights the interplay between seasonally variable atmospheric forcing and ice conditions, on a backdrop of long term changes to the Arctic sea ice-ocean system. Studies point to various mechanisms influencing the observed increase in Arctic Ocean surface stress, and hence geostrophic currents, in the '00s – e.g. decreased ice concentration/thickness, changing atmospheric forcing, changing ice pack morphology – however more work is needed to refine the representation of atmosphere-ice-ocean coupling in models before we can fully attribute causality to these increases.

# 1 Introduction

The mean surface circulation of the Arctic Ocean and surrounding seas is well established and a schematic is shown in Figure 1. In regions of sea ice cover, the Arctic surface circulation generally mirrors large scale patterns of ice drift, exhibiting two major surface circulation



features: the Beaufort gyre (BG) and the Transpolar drift. The BG is driven by the Beaufort Sea high pressure system, a semi-permanent feature of the Arctic atmospheric circulation, and is an anticyclonic circulation in the Canada Basin, transporting water and ice from the central Arctic to the Beaufort and Chukchi Seas (Proshutinsky et al., 2002). The Transpolar drift flows from the Russian Arctic toward Fram Strait and continues down the east coast of Greenland in the East Greenland Current, transporting fresh and cold surface water from the Arctic to the Nordic Seas (e.g., Aagaard and Carmack (1989), Proshutinsky and Johnson (1997)). This current continues down the east coast of Greenland, through Denmark Strait, south-westward and around the southern tip of Greenland. North Atlantic water enters the Norwegian Sea and flows north-eastward in the Norwegian Atlantic Current (Hansen and Østerhus, 2000; Orvik and Niiler, 2002; Nøst and Isachsen, 2003). The Norwegian Atlantic Current splits, the Barents Sea Branch entering the Barents Sea through the Barents Sea Opening and following the northern coast of Scandinavia and into the Kara Sea along the west coast of Novaya Zemlya and through the Kara Gate, the remainder flowing north and entering the Arctic in the West Spitsbergen Current before turning east along the northern Barents Sea shelf break (Schauer et al., 2002; Aagaard et al., 1987; Gammelrød et al., 2009; Aksenov et al., 2010).

Estimates of sea ice circulation have been produced at sub-monthly timescales using satellite ice feature tracking for almost forty years (as summarised by Sumata et al. (2014)). However, the ocean surface circulation variability at seasonal to decadal timescales is less well known, particular for regions of seasonal or perennial sea ice cover, owing to a lack of long-term and extensive observations under the ice. Direct measurements of ocean currents are limited to instruments mounted on moorings, ships and, more recently, Ice-Tethered Profilers (Cole et al., 2014). Surface geostrophic currents can be inferred from ocean dynamic height, which can be measured using hydrographic profiles of pressure, temperature and salinity, or by satellite altimetry. In the Arctic, estimates of dynamic height from hydrography are limited to long term means (due to lack of data coverage), and also to the deep basins where instruments can safely operate (McPhee, 2013). Specialised processing is required to estimate Dynamic Ocean Topography (DOT), and hence geostrophic





currents, from satellite altimetry over the ice-covered portion of the Arctic Ocean and hence previous estimates have been limited to long-term means (Farrell et al., 2012), intermittent seasonal means (Kwok and Morison, 2011) and, more recently, to the ice-covered portion of the Arctic only (Kwok and Morison, 2015; Mizobata et al., 2016). Time-variable satel-
5 lite gravimetry has been used to study circulation variability (Volkov and Landerer, 2013; Peralta-Ferriz et al., 2014), however this only captures variations in DOT due to ocean mass variability, missing the majority of DOT variability in the Arctic Ocean (Armitage et al., 2016). In this context, we calculate monthly geostrophic currents using satellite-derived estimates of DOT from the ice-covered and ice-free portions of the Arctic Ocean between 2003 and
10 2014, to create the longest record of extensive Arctic surface circulation to present.

The timespan covered by our data allows us to assess variability in Arctic surface circulation in the context of significant environmental change, particularly changing sea ice conditions. In polar regions, sea ice drift is largely driven by the action of the wind and the ocean surface circulation (Thorndike and Colony, 1982). The drag exerted by sea ice
15 on the ocean surface sets the upper ocean in motion, setting up Ekman currents and the transport of relatively fresh surface waters. The uneven distribution of freshwater causes horizontal salinity gradients, and in the surface layer this in turn tilts the DOT so that the ocean adjusts to geostrophic balance (McPhee, 2008). As such, changes in sea ice circulation are tightly coupled to upper ocean circulation. Arctic sea ice drift accelerated in
20 the '00s, and suggested causes include changing wind forcing, the reduction of sea ice concentration and thickness, and changing ice pack morphology which alters the coupling between the atmosphere and the sea ice (Ogi et al., 2008; Rampal et al., 2009; Spreen et al., 2011; Kwok et al., 2013; Olason and Notz, 2014; Tsamados et al., 2014; Martin et al., 2014, 2016; Petty et al., 2016). Meanwhile, observations suggest that ocean surface stress
25 increased during the '00s, particularly in the Beaufort Sea where there was an accumulation of liquid freshwater due to increased Ekman pumping (Proshutinsky et al., 2009; Giles et al., 2012; Krishfield et al., 2014) and increased geostrophic currents due to doming of the regional DOT (Giles et al., 2012; McPhee, 2013). Changes in ice circulation and ocean surface stress will influence the surface circulation and, likewise, changing ocean surface



circulation has a leading order effect on ice drift and ocean surface stress. Improved observations of upper ocean circulation will provide a better understanding of how this coupled system will evolve as sea ice retreats over the coming decades (Stroeve et al., 2012). We also examine the changing location of the Beaufort Gyre over the study period; the gyre is known to vary position along a northwest-southeast axis on decadal timescales (Proshutinsky et al., 2009) and we link observed changes in ocean geostrophic circulation to changes in the gyre location, and discuss implications for interactions between the gyre circulation and bathymetric features.

The oceanic kinetic energy is dominated by the mesoscale eddy field (Ferrari and Wunsch, 2008) and in the western Arctic Ocean eddies account for a significant proportion of the surface oceanic kinetic energy budget (Manley and Hunkins, 1985). Thus, as well as geostrophic currents, we also estimate seasonal climatologies of eddy kinetic energy (EKE), a metric of ocean variability that can be estimated from the variance of geostrophic current anomalies (Wunsch and Stammer, 1998). EKE has been estimated using satellite altimeters over the global ocean (Wunsch and Stammer, 1998) and in the Nordic Seas (Bulczak et al., 2015), and here we extend estimates of EKE into the central Arctic basins. There has been much recent interest in the role of eddies in the Arctic Ocean, particularly regarding their dissipative role in Beaufort Gyre dynamics (Timmermans et al., 2008; Cole et al., 2014; Manucharyan and Spall, 2016; Zhao et al., 2016), and we intend to provide a view of eddy activity that is complimentary to detailed in situ studies using profilers and modelling studies.

The paper is structured as follows: in section 2, we use the record of DOT to derive geostrophic currents; in section 3.1 we evaluate the satellite-derived currents against in situ data; in section 3.2 we characterise the seasonal to decadal variability of the Arctic Ocean geostrophic circulation; and in section 3.3 we analyse seasonal climatologies of EKE. In section 4 we place aspects of the seasonal to decadal variability in the context of changing Arctic environmental conditions.

## 2    Data and methods

We use the monthly Arctic DOT estimates from Envisat (2003-2011) and CryoSat-2 (2012-2014) produced by Armitage et al. (2016). These estimates of DOT combine sea surface height (SSH) from the open ocean and ice-covered ocean (via leads) to estimate basin-wide DOT up to 81.5°N (see Armitage et al. (2016) for full details of the data processing). To estimate monthly geostrophic currents the following steps are taken. Monthly pointwise DOT estimates are averaged into 0.75°×0.25° longitude-latitude grid, grid cells are masked if they are within 10km of land and we apply a Gaussian convolution filter with a standard deviation of 100km and a radius of 3 standard deviations to remove high frequency geoid undulations. In this study we completely mask the Canadian Arctic Archipelago, where the sparsity of data coverage and narrow straits results in noisy DOT estimates. The surface geostrophic current is given by:

$$\mathbf{u_g} = \frac{g}{f} \left( \hat{\mathbf{k}} \times \nabla_H \eta \right) \tag{1}$$

where $g$ is the gravitational acceleration, $f = 2\Omega \sin\phi$ is the Coriolis frequency, $\Omega$ is the rotation rate of the Earth, $\phi$ is the latitude, $\hat{\mathbf{k}}$ is the unit vector normal to the geoid, $\nabla_H = (\partial/\partial x, \partial/\partial y, 0)$ is the horizontal divergence operator and $\eta$ is the gridded DOT (Gill, 1982). Equation (1) represents the balance between the pressure gradient force and the Coriolis acceleration, under the assumption that the horizontal pressure gradient can be written as $\nabla_H p = \rho g \nabla_H \eta$.

We track the location of the BG by calculating the DOT centroid (i.e., the center of mass) as

$$\mathbf{r_c} = \frac{1}{\sum \eta_i} \sum \mathbf{r_i} \eta_i \tag{2}$$

where $\mathbf{r_i}(x_i, y_i)$ is the position of a given DOT grid cell. Before calculating the centroid, we mask all DOT grid cells over the shelf seas (<300 m depth) using the ETOPO1 global



bathymetry model (Amante and Eakins, 2009), and all grid cells outside of our BG region (the bounds of Figure 5). We also find the minimum closed DOT contour and use only the grid cells within this region, thus maximising the closed-contour area. This is similar to the approach of Proshutinsky and Johnson (1997) who use the lowest closed contour of DOT to define their Arctic Ocean Oscillation (AOO) Index. If no closed contours are found, this implies no obviously defined gyre and thus no monthly gyre centroid is produced (this occurs in a few instances in 2003/2004).

We estimate EKE by taking monthly geostrophic current anomalies, $\mathbf{u'_g}$, that are estimated from monthly DOT anomalies $\eta' = \eta - \bar{\eta}$ using equation (1), where the bar denotes the 2003-2014 time mean DOT. By subtracting the time mean DOT we remove the geoid height, which contains significant noise at high spatial frequencies, and less smoothing needs to be applied so the standard deviation of the Gaussian convolution filter is reduced to 25km. The surface EKE is then:

$$K_E = \frac{1}{2}\left(\langle u'^2 \rangle + \langle v'^2 \rangle\right) \tag{3}$$

where $\langle x \rangle$ denotes the time mean of $x$.

In section 3, we estimate seasonal fields of geostrophic currents and EKE by applying equations (1) and (3) to months with thick, consolidated ice conditions (November-June), referred to as 'winter', and to months with thin ice/ice free conditions (July-October), referred to as 'summer'. This allows us to look at seasonal variability and seasonal changes for broadly different surface forcing conditions; during summer the ice is more likely to be in a state of free drift and there will be more open water areas, but atmospheric circulation is weaker, and, during winter the ice pack will be more consolidated and internal stresses larger, but atmospheric circulation strongest.





## 3   Results

### 3.1   Data evaluation

We evaluate the satellite-derived geostrophic currents against in situ currents measured
by Acoustic Doppler Current Profilers (ADCPs) mounted on three moorings in the Beau-
fort Sea as part of the Beaufort Gyre Exploration Project (locations shown in Figure 1).
ADCPs have been attached to BGEP mooring D ($140°$W, $74°$N) since 2005 and to moor-
ings A ($150°$W, $75°$N) and B ($150°$W, $78°$N) since 2010 (mooring C was in place between
2005-2008, however no ADCP data is available). The ADCP is attached to the top of the
mooring at a depth of around 50m facing upwards, and profiles of the ocean current veloc-
ity are recorded every hour at 2m intervals. The current profile depth range varies between
instrument and between annual deployments, but is generally between 5–40m, with data
most reliably recorded over the 5–20m depth range (Figure 2, shaded regions). So, for con-
sistency, we calculate the mean eastward and northward current components in the upper
5–20m for each hourly profile. We find the monthly mean ADCP current components for
every month with more than 20 days of data available, and for each mooring location we
also produce a time series of the satellite-derived geostrophic current components, inter-
polated to the mooring location. Finally, we calculate the monthly mean current speed (i.e.,
$|\mathbf{u}| = \sqrt{u^2 + v^2}$) and bearing (degrees clockwise of north) for both the ADCP and satellite-
derived currents (Figure 2). The choice of the 5–20m depth range is influenced by the fact
that, in theory, the satellite-derived geostrophic current should best represent the current at
the surface so we use a shallow depth range that still allows us to perform a reasonable
amount of averaging, which is required to minimise the effect of small-scale velocity fluctua-
tions. We note that we have also performed the intercomparison by averaging over a variety
of different ADCP depth ranges (not shown), and whilst we find that it makes little difference
to the results, the best agreement is reached over the 5–20m depth range.

The in situ and satellite-derived currents show varying levels of agreement. Mooring D,
the longest record, shows the best correlation with the satellite data, with R = 0.54 for the
current speed and R = 0.35 for the current bearing ($p < 0.002$ in both cases). There were

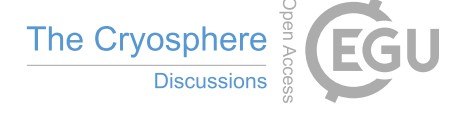

long-term changes in the current speed and bearing at this location over the course of the record; the current speed increased at this location in 2007-2008 and the satellite and in situ data show a drift in the current bearing from southwards in 2008-2009 to south-westwards by 2014. The only other significant correlation (at the $p < 0.05$ level) is found with the current speed at mooring B. At mooring A, the monthly mean current speeds of only $\sim$1cm s$^{-1}$ are at the same level as the root-mean-square (RMS) variability between the ADCP and satellite-derived currents, possibly explaining the lack of significant correlation at this location. In general the ADCP derived currents show more variability at monthly timescales than the satellite data, as reflected by the RMS variability and the low overall fraction of the variance explained (Table 1). This can at least partially be explained by considering that the effective footprint of the satellite currents is $\sim$100km owing to the smoothing function applied to reduce residual noise from the geoid, whereas the ADCP data are point measurements. Also, the ADCPs are measuring the actual current speed whereas the satellite data are used to estimate geostrophic currents and will not detect Ekman currents, so will be relatively insensitive to short temporal and spatial scale fluctuations due to eddies and ice motion. Despite the greater variability apparent in the ADCP currents, the mean difference between the ADCP and satellite-derived current speed and bearing is relatively low for all three records (Table 1), even for the poorly correlated records from moorings A and B. The ADCP current bearing measured at mooring B perhaps shows evidence of offsets between different annual deployments, which generally begin in August or September (Figure 2b). The current bearing during each deployment generally remains quite steady at this location, but varies between 100° in 2010-2011, 160° in 2011-2012, 30° in 2012-2013, and 150° in 2013-2014. The small mean differences in current speed and bearing give us some confidence that the satellite-derived geostrophic currents provide a reasonable representation of near-surface circulation at monthly timescales in the Beaufort Sea. Meanwhile, the RMS differences of $\sim$1.3cm s$^{-1}$ and $\sim$60° in current speed and bearing provide insight into the variability not captured by the satellite data. As far as we are aware, this represents the first direct comparison between satellite derived currents and in situ currents for the Arctic Ocean, despite the reporting of satellite-derived currents in the literature (e.g., Giles et al.





(2012); Morison et al. (2012); Kwok et al. (2013); Volkov and Landerer (2013); Peralta-Ferriz et al. (2014)).

## 3.2 Seasonal geostrophic currents and their variability

Seasonal mean geostrophic currents for 2003-2014 (Figure 3a–b) resemble the well-known surface circulation of the Arctic Ocean and surrounding seas (Figure 1). Some aspects of the mean circulation, principally the transpolar drift, are not resolved due to the latitudinal limit of Envisat (3a–b), however these features are well-resolved in the 2011-2014 mean DOT from CryoSat-2 which has coverage up to 88°N (Figure S1). The West Spitsbergen Current is not fully resolved in Figures 3a–b as it is only ∼100km wide (Beszczynska-Möller et al., 2012) and the Gaussian smoothing function applied to the DOT has a standard deviation of 100km (Armitage et al., 2016).

We calculate seasonal circulation anomalies relative to the seasonal means for three 4-year periods: 2003-2006, 2007-2010 and 2011-2014 (Figure 3c–h). We chose these time periods because the DOT in the Beaufort Sea showed a marked increase between 2006-2008 before decreasing slightly after 2011 (Armitage et al., 2016), and we wish to investigate the impact of these changes on geostrophic currents. To quantify these changes we examine time series of the monthly mean geostrophic current speed normal to three gates in the Beaufort Sea, chosen to characterise important aspects of the BG circulation: 1) between (129°W, 70.5°N) and (135°W, 72.5°N) in the southeastern Beaufort Sea, 2) between (163.5°W, 73.25°N) and (158.25°W, 75°N) in the southwestern Beaufort Sea and 3) between (155°W, 78°N) and (155°W, 80°N) in the northern Beaufort Sea. In addition, we examine the north-south current speed through Fram Strait, across 80°N between 12°W–0°E (Figure 4).

Geostrophic current speeds increased across almost the entire basin in both seasons between 2003-2006 and 2007-2010, with the most pronounced changes occurring in the Beaufort Sea (Figure 3e–f). The increase in Beaufort Sea DOT in late 2007 (Armitage et al., 2016) coincided with a peak in ocean surface slope and a maximum monthly mean current speed of 11.9cm s$^{-1}$ through the southeastern Beaufort Sea gate in November



2007 (Figure 4a). The changes between 2003-2006 and 2007-2010 are consistent with McPhee (2013), who estimated a 5–6 fold increase in current strength along the Beaufort and Chukchi shelf slopes between 2008-2011 relative to a climatology. By 2011-2014 the geostrophic currents in the southeastern portion of the BG had broadly returned to speeds seen in 2003-2006 (Figure 3g–h), with the slowest current speeds through the southeastern gate seen in 2012-2013, and a minimum current speed of -0.2cm s$^{-1}$ in November 2012 (Figure 4a). Meanwhile, there was a sustained increase in the current in the southwestern portion of the gyre (Figure 3g–h) and the annual mean current speed through the southwestern Beaufort Sea gate more than doubled, from 2.8cm s$^{-1}$ in 2003 to 6.0cm s$^{-1}$ in 2014 (Figure 4b).

The overall pattern of change between 2007-2010 and 2011-2014 shows the BG circulation shifting to the west, with the southeast portion of the gyre slowing and stronger currents in the western portion of the gyre (Figure 3g–h). This is confirmed by the changing location of BG centroid (Figure 5): between 2003 and 2014 the gyre centroid drifted north west, from the central Beaufort Sea ($\sim$145°W, 74°N) in 2003, to the northwestern Beaufort Sea, adjacent to the Northwind Ridge ($\sim$150°W, 76°N). The maximum deviation in the annual mean gyre position was seen in 2013, when the gyre centroid was more than 300km northwest of its 2003 position. There was a small increase in the eastward flow in the northern portion of the BG in 2008 (Figure 4c).

The southward current through Fram Strait and the East Greenland Current strengthened between the periods 2003-2006 and 2007-2010, and this was mostly maintained in 2011-2014 (Figure 3c-f). The annual mean current through the Fram Strait gate increased from 4.6cm s$^{-1}$ in 2003 to a maximum of 7.0cm s$^{-1}$ in 2011 and 2012, before slowing to 5.1cm s$^{-1}$ in 2014 (Figure 4d). From the increase in the Fram Strait current and the northward currents feeding the transpolar drift, we infer that there was likely a strengthening of the transpolar drift between 2003-2006 and 2007-2010. This inference is supported by the current speed recorded on the North Pole Environmental Observatory mooring between 2003 and 2010, which saw a doubling of the transpolar drift current speed in 2008 (Figure S2).



The geostrophic circulation is strongest in winter and weakest in summer (Figure 3a–b). The summertime BG circulation was weak between 2003-2006, but it became a prominent feature of the summertime circulation in later periods. There is a distinct seasonal cycle in the current through the southeastern Beaufort Sea gate that broadly peaks between October and December and is weakest in August, with a seasonal range of 2.8cm s$^{-1}$ and large interannual variability (Figure 6a). There is also a seasonal cycle in the Fram Strait current that is strongest in December and weakest in May, August and September, with a seasonal range of 1.9cm s$^{-1}$ (Figure 6b).

## 3.3 Seasonal eddy kinetic energy

Seasonal climatologies of EKE show considerable spatial inhomogeneity, revealing higher eddy activity in winter than in summer (Figure 7). We find coherent, year-round features of the EKE field, apparently controlled by the interaction between bathymetry, the sea ice edge and the mean flow. Bulczak et al. (2015) reported on EKE variability in the Nordic Seas and we note only that our data corroborates their observations (their data was a subset of the data used by Armitage et al. (2016)). The East Greenland Current is known to be abundant with eddies (e.g., Foldvik et al. (1988)) associated with the interaction between the strong current, the ice edge and the east Greenland shelf, with high EKE generally concurring with the shelf break (the 1km isobath contour). This band of high EKE continues around southern Greenland, following the cyclonic current in the Labrador Sea, with a region of eddy activity downstream of Cape Desolation (Eden and Bóning, 2002). There is high EKE associated with west Greenland and Baffin Island currents, despite these features being poorly resolved in Figure 3a–b, and a persistent area of high EKE extending into Baffin Bay around 73°N. The Barents Sea shows relatively high EKE, particularly in winter, with a hotspot of eddy activity on the western periphery of the Svalbard Bank that has also been observed by drifters (Loeng and Sætre, 2001). It has been known for some time that eddies are prevalent in the western Arctic Ocean, and that they make up a significant portion of the oceanic kinetic energy budget in this region (Manley and Hunkins, 1985). We find high EKE



along the topographic margins of the Beaufort Gyre extending along the Northwind Ridge and Chukchi Ridge formations, consistent with the findings of Zhao et al. (2016).

We only present seasonal climatologies of EKE as the monthly fields are noisy, particularly in ice-covered regions and where uncertainties due to poor geophysical corrections are greater. Background levels of EKE in the ice-covered Arctic are influenced by the spatio-temporal sampling of the surface as evidenced by 'trackiness' in the EKE fields and lower EKE adjacent to the 'pole hole' where the spatial density of ground tracks increases. High RMS noise in the SSH data due to poor tidal corrections shoreward of the 50m isobath in the East Siberian, Laptev and Bering Seas dominates over any signal related to eddy activity. Furthermore, in these shallow shelf seas, the Rossby radius is just 1–2km (Nurser and Bacon, 2014), so SSH deflections associated with these eddies are not likely to be detectable from altimetry (∼300m along track sampling). This does not apply in the deeper Greenland continental shelf and Labrador Sea regions, where the RMS noise is lower.

## 4   Discussion

There was a confluence of anomalous environmental conditions in the second half of 2007 that contributed to the intensification of currents in the Beaufort Gyre in late 2007. Strong and persistent high pressure anomalies in the Beaufort Sea drove strong anticyclonic winds (Figure S3), there was a record low sea ice extent in September 2007, a significant loss of multi-year ice in the Beaufort Sea (Maslanik et al., 2011), and the ice pack was significantly thinner in the 2007-2008 growth season than the preceding five years (Giles et al., 2008). This meant that in autumn 2007, a weaker and more mobile ice pack could be driven more efficiently by intense and persistent anticyclonic winds (Petty et al., 2016), driving strong Ekman convergence in the BG (Proshutinsky et al., 2009) and doming of the DOT (Giles et al., 2012; McPhee, 2013; Armitage et al., 2016). The extreme slope in BG DOT was registered as large drops in SSH by tide gauges on the periphery of the gyre, the Tuk-toyaktuk tide gauge in northern Canada recording a −50cm SSH anomaly in November 2007 (Armitage et al. (2016), their Supplementary Information 2). This resulted in strong

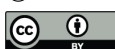

geostrophic currents around the BG in late 2007 that were, in the southeastern portion of the gyre, anomalous between 2003 and 2014 (Figure 4).

The BG circulation remained elevated throughout 2007-2010. This period saw enhanced ice circulation around the gyre that was partially linked to enhanced atmospheric circulation (Petty et al., 2016) but also changes in the sea ice characteristics, e.g. the loss of multiyear ice cover (Maslanik et al., 2011), and the increased efficiency of momentum transfer between the atmosphere and ocean (Giles et al., 2012; Martin et al., 2014; Petty et al., 2016). Stronger currents in the southern and western gyre likely also contributed to increased advection of older, thicker ice westwards toward the Chukchi and Siberian shelf seas where it is more easily melted during summer (Hutchings and Rigor, 2012) or can join the transpolar drift and exit the Arctic Ocean through Fram Strait (McPhee, 2013; Carmack et al., 2015). Currents in the southeastearn and northern BG remained elevated until around 2011-2012, before reducing to values representative of 2003-2006 (Figure 4a,c). As discussed earlier, between 2011-2014 the BG also shifted to the northwest (Figure 3g–h; Figure 5), with the northwestward current through the southwest Beaufort Sea gate remaining elevated (Figure 4b). The BG is known to shift position at decadal timescales in response to varying atmospheric forcing; the BG centroid drift from ($\sim$145°W, 74°N) to ($\sim$150°W, 76°N) over the period of this study is consistent with the BG moving from its typical 1990-2000s location back to a position more representative of the 1950s-1980s (Proshutinsky et al., 2009).

Proshutinsky and Johnson (1997) defined the Arctic Ocean Oscillation (AOO) index, based on the DOT slope in the central Arctic basin, to characterise the Arctic ocean circulation regime. The AOO has been in an anticyclonic phase since the 1990s, characterised by freshwater accumulation and expansion of the BG, but since 2011 the anticyclonic AOO index has weakened (Proshutinsky et al., 2015). Petty et al. (2016) reported a reduction of wind curl (a proxy for Ekman pumping) in the Beaufort Gyre region since 2010. Our data showing doming of the BG which reflects freshwater accumulation (Armitage et al., 2016) and enhanced circulation up to 2010, with a release of freshwater (Armitage et al., 2016) and a relaxing of the oceanic circulation since 2010-2011. It remains to be seen whether this signals a phase change in the AOO index, or simply interannual variability.





The major circulation features of the Arctic Ocean and surrounding seas are stronger in the winter than summer (Figure 3a–b). The strength of the BG circulation and East Greenland Current are modulated by the seasonal intensity of the Beaufort Sea high and Icelandic low pressure systems (Proshutinsky et al., 2002; Serreze and Barrett, 2011; Bacon et al., 2014). Despite a more compacted ice pack and higher ice interaction forces between November and June (which tend to oppose ice motion and dampen momentum transfer to the ocean), wintertime atmospheric forcing is sufficiently strong to result in higher ocean surface stress and geostrophic currents. The seasonally varying mean circulation interacts with seafloor bathymetry to drive seasonal variations in EKE, which is also higher in winter than summer (Figure 7). This is particularly evident over the Northwind Ridge and Chukchi Plateau formations, where the BG circulation is strongest and intersects steep topographic features (Figure 3a–b and Figure 7). Mesoscale eddies are thought to play an important role in the balance between storage and release of freshwater in the BG (Manucharyan and Spall, 2016). Freshwater accumulation and steepening of isopycnals over a timescale of years causes a build up of the potential energy, which is dissipated by eddies until the gyre reaches a steady state. Zhao et al. (2016) observed enhanced eddy activity in the western Beaufort Sea after 2012, and linked this to dissipation of the potential energy built up by freshwater storage in the late '00s. It is also possible that the northwest drift of the BG centroid (Figure 5) is also implicated in the enhanced eddy activity observed by Zhao et al. (2016), as a larger portion of the gyre circulation intercepts the Northwind Ridge and Chukchi Plateau. The EKE fields produced by altimetry are noisy even at interannual timescales meaning we do not resolve interannual variability, however we are able to average enough data to produce seasonal climatologies (Figure 7), offering a complimentary view to the hydrographic data presented by Zhao et al. (2016) who do not resolve seasonal variability. Zhao et al. (2016) note that observations of eddies are important for comparison with ocean models as they become eddy-resolving in the Arctic Ocean.

Whilst the wintertime circulation is stronger on average, the increase in circulation between 2003-2006 and 2007-2010 was larger in summer than in winter (Figure 3c–f). This highlights the complex interplay between seasonal differences in wind forcing and seasonal

changes in ice conditions and atmosphere-ice-ocean coupling. Modelling studies provide some insight into how different factors affecting atmosphere-ice-ocean momentum fluxes have contributed to seasonal changes in ocean surface stress and geostrophic currents. Summertime circulation changes between 2003-2006 and 2007-2010 are likely a result of increased coupling between the atmosphere and the ocean due to reduced ice concentration. Tsamados et al. (2014) found increases in the modelled ice-ocean drag coefficient, largely resulting from increased floe edges due to a less concentrated ice pack. Martin et al. (2014) found that, in the '00s, the summertime ice pack was experiencing longer periods of 'optimal ice concentration' (80–90% ice concentration) for momentum transfer to the ocean. However, in this study, and a follow-up study including variable form drag (Martin et al., 2016), they reported a negative trend in summertime ocean surface stress; despite increased ice-ocean stress, the loss of summer sea ice coverage led to an overall decrease in the ocean surface stress because the atmosphere-ocean stress is smaller than ice-ocean stress. Wintertime circulation changes are likely a result of reduced ice strength and reduced ice interaction forces. Ice strength is a strong function of ice thickness and concentration. Reductions in Beaufort Sea ice concentration in most seasons (apart from January-March) and thinning of the ice pack has likely reduced the ice interaction force (Petty et al., 2016) and, with less opposition to drift, the ice has drifted faster (Spreen et al., 2011) leading to an increased ocean surface stress (Martin et al., 2014). However, Martin et al. (2016) found reduced wintertime ocean surface stress due ice smoothing and decreased form drag associated with loss of deformed thick ice. So, whilst observations make it clear that ocean surface stress increased in the Arctic Ocean, particularly in the late '00s, contradictory model results show that more work is needed to refine the representation of atmosphere-ice-ocean coupling in models before we can fully attribute causality to these increases (Martin et al., 2016). An aspect of ice-ocean coupling that has been lacking is long term observations of time variable upper ocean circulation, which this study has helped to provide.



# 5   Conclusions

We have presented a new 12-year record of geostrophic circulation in the ice-covered and ice-free Arctic Ocean south of 81.5°N, the most extensive and longest record of Arctic surface circulation to date. Arctic Ocean geostrophic currents exhibit seasonal, interannual and decadal variability. The BG circulation accelerated in autumn 2007 when strong atmospheric circulation, record low sea ice extent, loss of multiyear ice and significant ice pack thinning resulted in high surface stress and spin-up of the ocean currents. Increases in circulation between 2007-2010 relative to 2003-2006 were larger in summer than in winter. Higher current speeds were sustained in the southeastern BG until 2011-2012 after which they relaxed to 2003-2006 levels, whilst the southwestern portion of the gyre shows a longer term increase, more than doubling between 2003 and 2014. This overall pattern of changing currents in the Beaufort Gyre during this time period is linked to a shifting of the gyre, and the associated ocean circulation features, to the northwest. In 2013 the gyre centroid was more than 300km to the northwest of its 2003 position. Fram Start current increased between 2003 and 2012, before slowing somewhat by the end of the time period. Both the BG and Fram Strait currents, dominant features of the upper Arctic Ocean circulation, show seasonal cycles (strongest in winter) that are modulated by the seasonal intensity of the Beaufort Sea high and Icelandic low pressure systems. Seasonal climatologies of EKE show persistent features associated with the interaction between the mean flow, bathymetric features and the sea ice edge, and EKE is higher in the winter across the whole Arctic due to the stronger wintertime currents. Our data provide the most detailed view yet of surface circulation changes in the Arctic in a period of significant Arctic environmental change.

*Acknowledgements.* This work was supported by the Natural Environment Research Council (grant number NE/M015238/1). The contribution of SB was supported by the NERC Arctic Research Programme TEA-COSI project (http://teacosi.org/; grant number NE/I028947/1). Radar altimetry data were supplied by the European Space Agency (https://earth.esa.int/web/guest/data-access/). ADCP data were collected and made available by the Beaufort Gyre Exploration Program based at the Woods Hole Oceanographic Institution (http://www.whoi.edu/beaufortgyre) in collaboration with re-





searchers from Fisheries and Oceans Canada at the Institute of Ocean Sciences, and North Pole Environmental Observatory ADCP data is available via http://psc.apl.washington.edu/northpole/index.html.

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

**Table 1.** Comparison between the satellite derived geostrophic currents and in situ currents measured at three BGEP moorings.

| | Mooring A | | Mooring B | | Mooring D | |
|---|---|---|---|---|---|---|
| | Speed | Bearing | Speed | Bearing | Speed | Bearing |
| $N_{months}$ | 38 | | 50 | | 79 | |
| Correlation $(R^2)^*$ | 0.14 | 0.05 | **0.34** | 0.07 | **0.54** | **0.35** |
| | (0.02) | (0.00) | **(0.11)** | (0.01) | **(0.29)** | **(0.12)** |
| Mean difference | 0.5cm s$^{-1}$ | -6° | -0.3cm s$^{-1}$ | 15° | -0.2cm s$^{-1}$ | -17° |
| RMS difference | 1.0cm s$^{-1}$ | 67° | 1.7cm s$^{-1}$ | 62° | 1.1cm s$^{-1}$ | 53° |

*Correlations significant at the $(p < 0.05)$ level are shown in bold.

Discussion Paper | Discussion Paper | Discussion Paper | Discussion Paper







**Figure 1.** Map of the Arctic Ocean and surrounding seas with a schematic of the ocean surface circulation (blue arrows). Depth contours are drawn every 1km and are taken from the ETOPO1 global bathymetry model (Amante and Eakins, 2009). Bathymetric features mentioned in the text are labelled in red, with the following abbreviations: BS = Bering Strait, NR = Northwind Ridge, CP = Chukchi Plateau, FS = Fram Strait, BSO = Barents Sea Opening, SB = Svalbard Bank, KG = Kara Gate, LB = Lofoten Basin, DS = Denmark Strait. Specific currents mentioned in the text are labelled blue, with the following abbreviations: BG = Beaufort Gyre, TPC = Transpolar drift, WSC = West Spitsbergen Current, EGC = East Greenland Current, WGC = West Greenland Current, BIC = Baffin Island Current, NAC = Norwegian Atlantic Current, BSB = Barents Sea Branch. Beaufort Gyre Exploration Project moorings A, B and D are labelled in white.

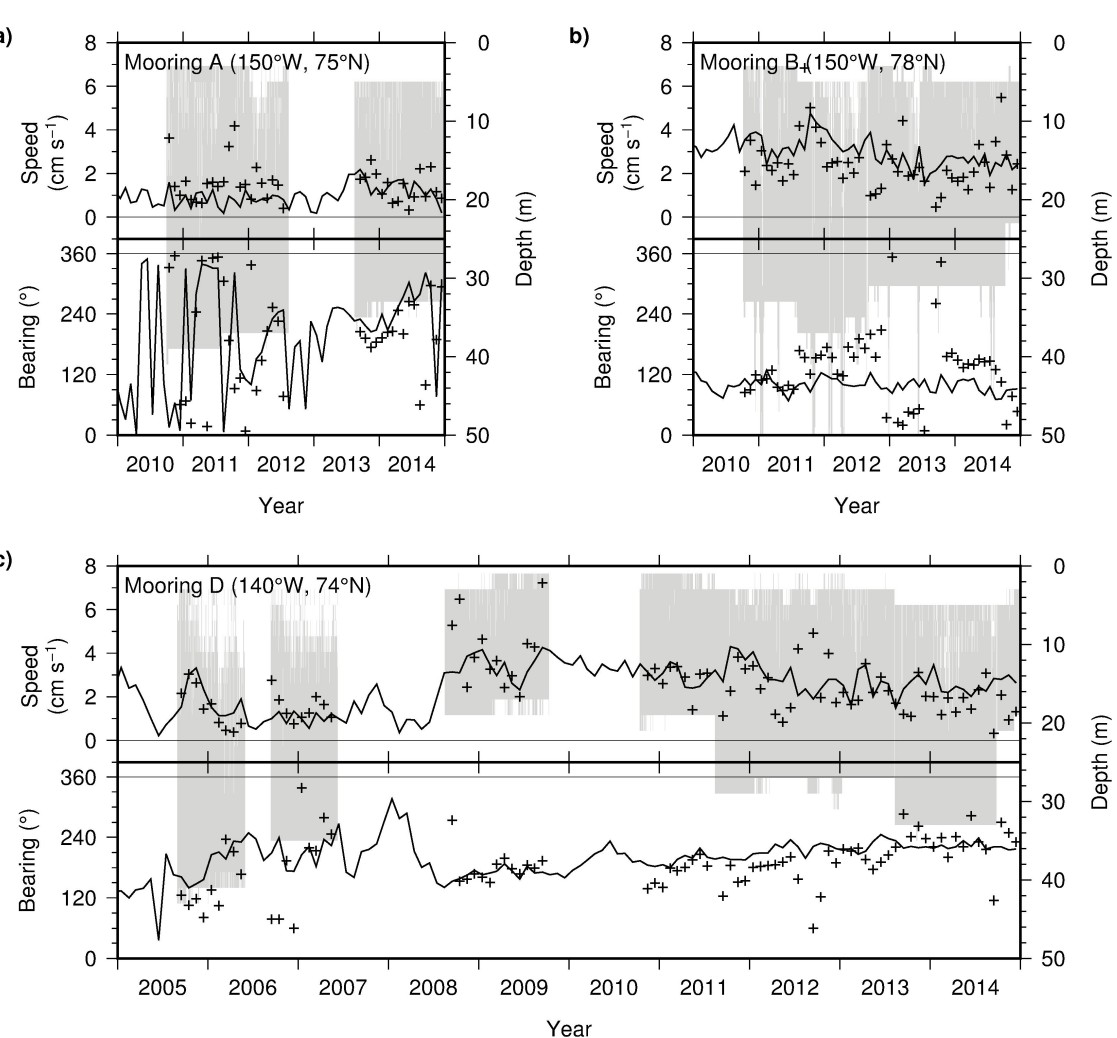

**Figure 2.** The satellite-derived current speed and bearing (black lines) are shown against the ADCP-derived currents (crosses) for a) mooring A, b) mooring B and c) mooring D. The data availability is shown as a function of depth (right axes) by the grey shaded regions.

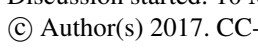



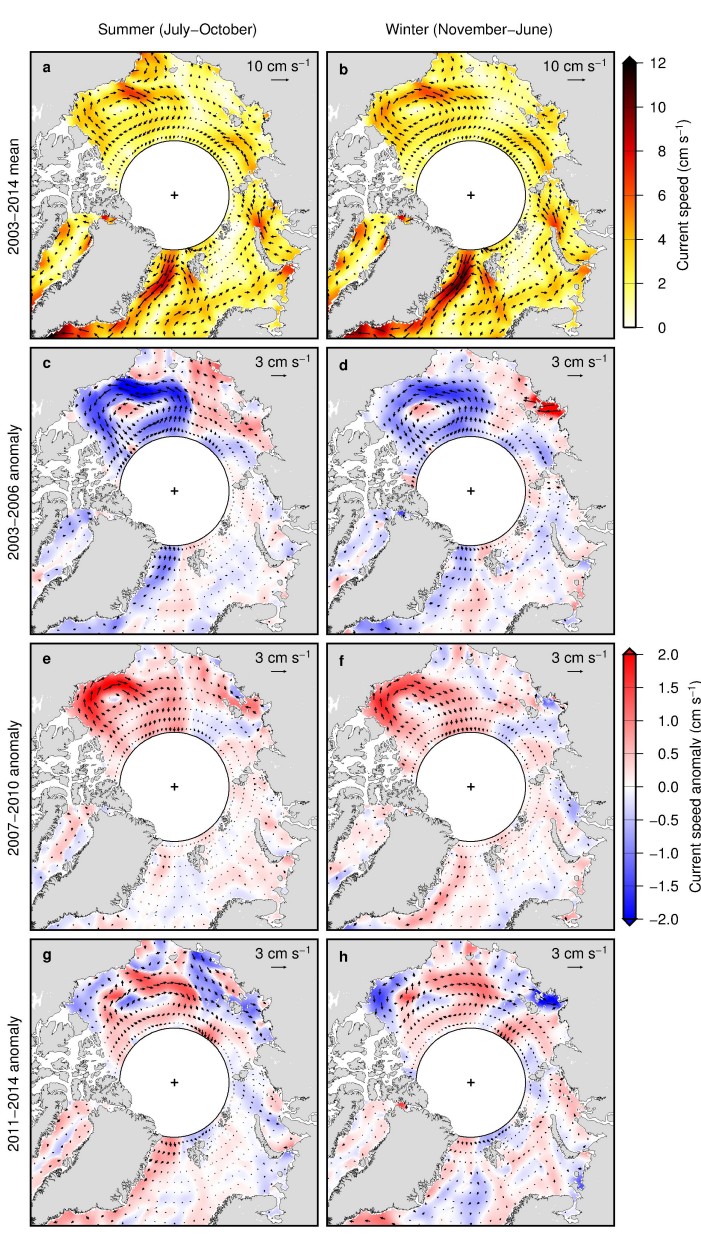



**Figure 3.** The 2003-2014 July-October (left column) and November-June (right column) mean geostrophic currents (a–b) and the current anomalies in successive four year periods, from top to bottom: 2003-2006 (c–d), 2007-2010 (e–f) and 2011-2014 (g–h).





**Figure 4.** The geostrophic current speed normal to various gates in the Arctic Ocean: the colour of the time series corresponds to the colour of the gate on the map (right). The time series, from top to bottom, are the current speed through the southeastern (positive southwest), southwestern (positive northwest) and northern (positive east) gates in the Beaufort Sea and the Fram Strait gate (positive south). The 2003-2014 mean geostrophic velocity vectors are shown on the map. The y-axis has been scaled such that the vertical increment is constant.

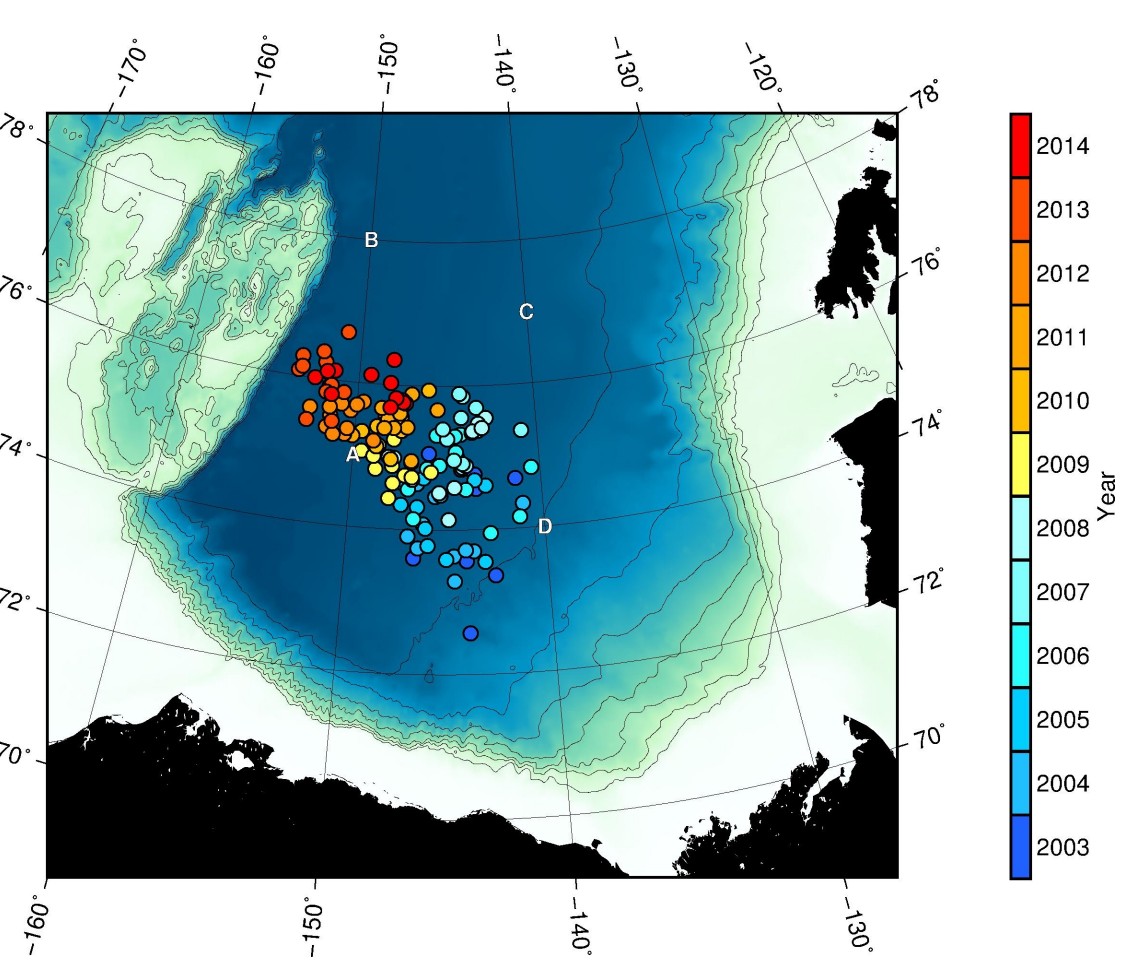

**Figure 5.** The monthly Beaufort gyre centroid location estimated from the DOT. Depth contours are drawn at 1km intervals, taken from the ETOPO1 global bathymetry model (Amante and Eakins, 2009). The Beaufort Gyre Exploration Project mooring locations are shown in white.

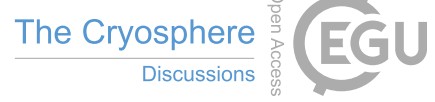



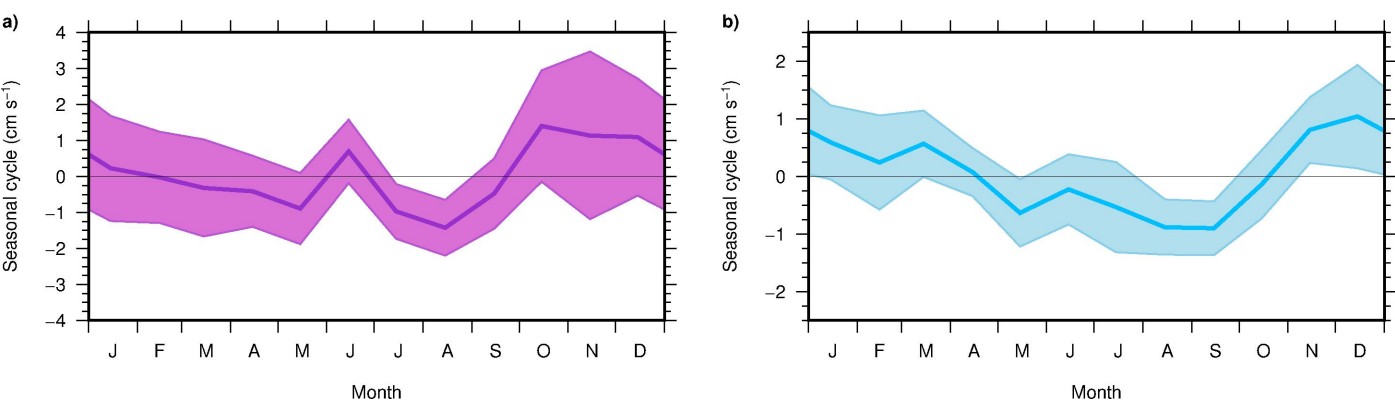

**Figure 6.** The seasonal cycle in the geostrophic currents a) normal to the southeastern Beaufort Sea gate (positive southeasterly) and b)the north-south geostrophic currents through Fram Strait (positive south). The annual mean current is removed from each year of data and we estimate the monthly mean and standard deviation (shown by the shaded band) of the resulting seasonal anomalies. The colour corresponds to the gate colour shown in Figure 4.

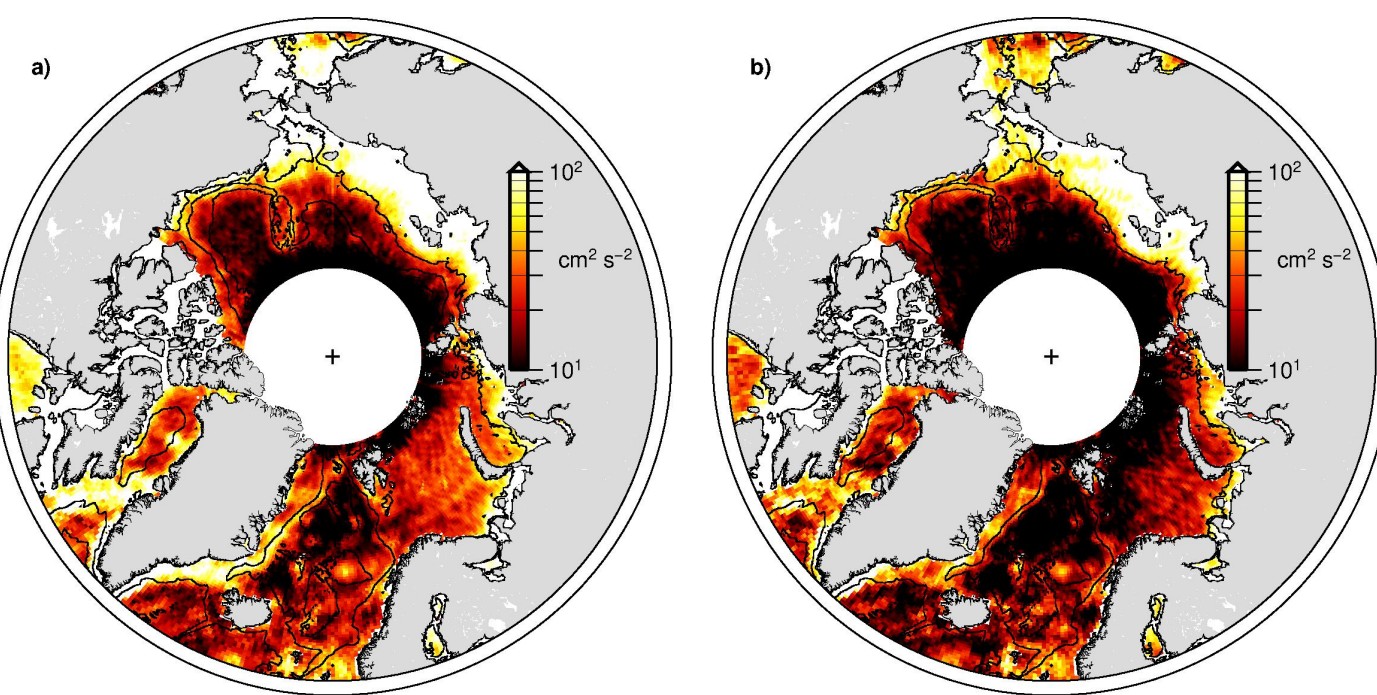

**Figure 7.** The November-June (a) and July-October (b) eddy kinetic energy. Depth contours are drawn at 50m, 1km and 3km, taken from the ETOPO1 global bathymetry model (Amante and Eakins, 2009). Note that the Canadian Arctic Archipelago has been masked out in these plots (see section 2).