# Peer review of "Manuscript prepared for The Cryosphere Discuss. with version 2015/09/17 7.94 Copernicus papers of the LATEX class copernicus.cls. Date: 24 June 2017"

_The Cryosphere, 2017_

## Short Comment (SC1) · 24 Mar 2017

I would like to make a comment on the importance of the derived eddy kinetic energy (EKE).

While capturing only large-scale eddies, the EKE estimates provide first-of-a-kind global evidence of eddy variability on a gyre-scale, which makes it a crucial dataset for the advancement of our understanding of BG and freshwater variability.

A recently developed theory explicitly links FWC, halocline depth, and geostrophic currents to eddy dynamics. In particular, Manucharyan and Spall 2016 suggest that lateral freshwater fluxes due to eddies are counteracting the Ekman-driven freshwater accumulation. As a result, a characteristic isopycnal slope, s, that is linearly proportional to geostrophic currents and should scale as

s $\sim$ tau / (rho f K), (1)

where tau is the surface stress, rho - density of the ocean, f - Coriolis parameter, and K is the isopycnal eddy diffusivity. Idealized BG simulations suggest that a realistic halocline can be achieved if K is in a range of 100–500m^2/s, with lower values in the interior of the gyre and higher values near its coastal boundaries (see Figure 3a in Manucharyan et al, 2016). However, due to the scarcity of data, we currently lack global observational evidence to confirm these values of diffusivities, thus leaving the theory as a hypothesis.

Nonetheless, eddy diffusivity parameter K can be estimated based on a mixing length theory that has been tested in other world oceans (Holloway, 1986). In particular, Klocker and Abernathey (2014) suggest an unsuppressed eddy diffusivity can be calculated as

K=gamma u_rms Lmix, (2)

where Lmix is the mixing length that is of the order of the Rossby deformation radius Rd, the characteristic eddy velocity u_rms is taken as u_rms=sqrt(EKE), and gamma$\sim$0.35 is an empirically estimated efficiency coefficient that stays nearly constant for a wide range of flows. For the sake of making a rough estimate the diffusivity (and compare it to idealized simulations), (2) can be rewritten as

K=0.35 sqrt(EKE) Rd. (3)

Estimating sqrt(EKE) at about 0.1 m/s near coastal boundaries and about 0.05 m/s in the interior of the gyre (Figure 7 of the manuscript under review), we find that these values are consistent with idealizes BG simulations (see Figure 2b in Manucharyan et al, 2017). Taking Rd = 15 km in the BG (Nurser and Bacon, 2014), and using Eq. (3) we get the following range for K = 250–500 m^2/s which are also consistent with idealized BG simulations of Manucharyan et al, 2016 (see Figure 3a).

Note, that because of the limitation of constructing under-ice SSH data, the satellite

EKE estimates capture only large scale eddies. However, in the Arctic Ocean, a significant inverse cascade is expected to occur based on f-plain geostrophic turbulence theory (Larichev and Held, 1995). It is also expected that eddies with scales much larger than the local deformation radius should be dominant contributing to eddy buoyancy fluxes (see Figure 1 in Larichev and Held, 1995). This view is consistent with the idealized simulations of Manucharyan and Spall (2016) that resulted in eddies that are about 100km in scale.

In conclusion, the satellite EKE and K estimates based on large-scale variability could be adequate, and present a foundation for adjusting these values in climate models to improve the mean state and variability of the Beaufort Gyre.

It would be beneficial for the Arctic observational and modeling community if the authors comment in their manuscript on the relation between geostrophic currents and eddy dynamics.

G.E. Manucharyan

——- References:

Nurser, A. J. G., & Bacon, S. (2014). The Rossby radius in the Arctic Ocean. Ocean Science, 10(6), 967-975.

Klocker, A. and R. Abernathey, 2014: Global Patterns of Mesoscale Eddy Properties and Diffusivities. J. Phys. Oceanogr., 44, 1030–1046, doi: 10.1175/JPO-D-13-0159.1.

Manucharyan G.E. and M.A. Spall (2016), Wind-driven freshwater buildup and release in the Beaufort Gyre constrained by mesoscale eddies, Geophysical Research Letters, 43(1), pp 273-282.

Manucharyan, G., M. Spall, and A. Thompson, 2016: A Theory of the Wind-Driven Beaufort Gyre Variability. J. Phys. Oceanogr., 46, 3263–3278, doi: 10.1175/JPO-D-16-0091.1.

Manucharyan, G., A. Thompson, and M. Spall, (2017). Eddy-Memory mode of multi-decadal variability in residual-mean ocean circulations with application to the Beaufort Gyre. J. Phys. Oceanogr., 0, doi: 10.1175/JPO-D-16-0194.1.

Larichev, V. D., & Held, I. M. (1995). Eddy amplitudes and fluxes in a homogeneous model of fully developed baroclinic instability. Journal of physical oceanography, 25(10), 2285-2297.

Holloway, G. (1986). Estimation of oceanic eddy transports from satellite altimetry. Nature, 323(6085), 243-244.

---

## Referee Comment (RC1) · Anonymous Referee #1 · 19 Apr 2017

I enjoyed reading this paper which is well written and clearly set out. It describes the geostrophic circulation of the Arctic over the past 10 years or so derived from thermal wind calculations based on dynamic height estimates given in Armitage et al (2016). It is not clear to me the extent to which the study moves the field forward – I am not an Arctic expert and it is difficult for me to judge. The study is rather descriptive, however, and needs to be firmed up.

I have some suggestions that make help to make the paper more substantive.

1. How do the analysis presented here (and in Armitage, 2016) compare with other state estimates of the Arctic such as MIMOC, WOA, PHC or the NCEI climatology, or. eg., ocean data assimilative products? Do other analyses lead to essentially the same

circulation, T, S, ice pattern changes over the last decade? Is it possible to estimate the error bars or uncertainty on your estimates somehow?

2. The paper is very descriptive and there is little discussion of underlying mechanisms. Can the authors delve a little deeper and do more than describe what is happening and reporting that 'so-and-so said this' etc. At the moment the paper is very value-neutral and not very deep. It would be good to have some viewpoint expressed.

3. There has been considerable effort by the community to measure time-series across key Straits - Fram, Davis etc. Why not compare the estimates derived here with those that are directly measured? Is it because one needs the Ekman transport too? – but that would be directly and readily given by the wind, mediated by ice. Instead we are presented with geostrophic transports through the mini-sections shown in Fig.4, sections that have not been instrumented and thus not directly measured. This is a rather serious shortcoming of the paper I believe.

4. The paper would benefit from a plot of timeseries of key metrics such as FWC, AOO, sea-ice extent etc etc, so that they can be compared with those presented by, e.g, Proshutinsky.

Fig.1 is key, could be a very useful figure, and yet not easy to read. The labels need to be more distinctive and the confluence of dark blue and dark grey is not easy to parse. Could this be redrawn with attention to labels, colors etc so that they can be more easily read.

Fig.7 also needs some attention. Many of the details cannot be discerned.

---

## Referee Comment (RC2) · Anonymous Referee #2 · 27 Apr 2017

This paper presents estimates of the surface geostrophic circulation over much of the low-latitude Arctic Ocean and Nordic Seas derived from satellite SSH. The focus is on seasonal and interannual variability from 2003 through 2014. It is shown that the Beaufort Gyre changed in strength and location on roughly pentadal time scales. It is suggested that these changes were forced by changes in wind stress and ice cover. I am of mixed feelings with this paper. There is nothing wrong with the analysis, but it is very descriptive and I did not feel that I learned very much from it. It is useful to document these types of observations, however, so I am in favor of publication subject to some relatively minor revisions as outlined below.

Title: Should really be "...surface geostrophic circulation 2003-2014"

Abstract: State what data is used to derive the velocity estimates.

Figure 1: Poor choice of color scale, can this be made easier to read?

(page 3, line 4) Water and ice are not transported from the BG into the Chukchi Sea, it is the other way around.

(4,1) Why can you do seasonal and interannual analysis while previous studies were limited to long term means?

(4,28) The authors are correct that the ocean and ice coupling can go both ways, unfortunately they do not attempt to decompose which is driving the other in their analysis.

(5,11) The Manley and Hunkins eddies are very small, O( 10 km), and thus not detectable in the present filtered satellite data. This is a very important distinction. The EKE estimated here is more likely dominated by the gyre instability, which has been linked to wind stress and freshwater content by Manucharyn and Spall (2015 GRL) and Manucharyan et al. (2016 JPO).

(9,2) Looks more like the speed increased from 2008-2009.

(9,26) Is the direction difference partially explained by Ekman dynamics?

Figure 3: The vectors are very difficult to interpret (too small). I suggest contouring DOT together with the color plot of the velocity magnitude.

(11,1) I find the "gate" approach ambiguous. How much of the changes are due to changes in magnitude versus changes in gyre location?

(11,14) Does the BG extend over the Northwind Ridge when its centroid is located to the extreme NW? This is a little surprising to me.

(12,1) Figs 3a, b look nearly the same to me. I would guess if one were to put error bars on these they would be the same.

(12,13) In what way does the present study corroborate Bulczak et al?

(12) The numerous features of EKE that are mentioned are difficult to locate on the

figure. Maybe some geographic indicators would help the reader. Again, the EKE found here is probably not the small Manley and Hunkins eddies generated from the shelf break. It would be interesting to try to connect the magnitude of this EKE to that predicted by Manucharyan's work.

(13,6) I do not see any trackiness in the figure.

Discusson: The discussion is really a summary of other people findings and a recap of the present results connected by some speculation. I don't think it contributes much to the paper as is.

(17,14) Fram Strait

---

## Author Comment (AC1) · 10 Jun 2017

We have attached our response to the reviewer comment as a supplement to this comment.

Please also note the supplement to this comment:
http://www.the-cryosphere-discuss.net/tc-2017-22/tc-2017-22-AC1-supplement.pdf

---

## Author Comment (AC2) · 10 Jun 2017

We have attached our response to the reviewer comments as a supplement to this comment.

Please also note the supplement to this comment:
http://www.the-cryosphere-discuss.net/tc-2017-22/tc-2017-22-AC2-supplement.pdf

---

## Author Comment (AC3) · 10 Jun 2017

**Response to reviewers: "Arctic Ocean surface geostrophic circulation 2003-2014"**

T. Armitage, S. Bacon, A. Ridout, A. Petty, S. Wolbach, M. Tsamados
June 2017

We would like to thank the reviewers for their constructive reviews of our manuscript. We have made some revisions to the manuscript where we believe they were warranted and where we have chosen not to make changes we have provided our reasoning in our responses to specific reviewer comments. The two anonymous reviewers both commented that the manuscript was 'very descriptive'. We would like to make the argument that this is not in fact a negative aspect of the paper. The purpose of the paper was indeed to describe (qualitatively and quantitatively) variability in Arctic Ocean surface geostrophic circulation that has not been seen before by any dataset over this time period at monthly to decadal time scales. We have attempted to place the new data within the context of the current literature and we have provided a discussion of how the observed changes fit with the broader picture of Arctic environmental change. We anticipate that the data will be useful for future studies of the physical mechanisms behind the observed changes and variability

**Summary of major changes**

1. We have changed the title to "Arctic Ocean **surface** geostrophic…"
2. We have added further discussion on the satellite derived EKE estimates based on the comment of G. Manucharyan (page 15 line 26 – page 16 line 27). We have acknowledged his contribution in the acknowledgements (page 19 lines 1-3).
3. We have updated Figures 1, 3 and 7 to make them easier to read.
4. We have added a description of the important EKE features in the Nordic Seas reported by Bulczak et al. (2015). (Page 12 lines 13-23)

**Response to the comments of G. Manuscharyan**

We would like to thank you for your excellent observations and for allowing us to utilise this comment directly in the revised manuscript, which we believe has significantly improved our discussion around the EKE estimates in the paper. In the revised manuscript we have included a comparison of rough empirical estimates of the eddy diffusivity based on the satellite data, with the values found through the theoretical analysis present in Manucharyan and Spall (2016) (page 15 line 26 – page 16 line 15). We have also included (also in response to the comments made by another reviewer) elements of your discussion about the current variability seen by the satellites compared to the local radius of deformation (page 16 line 16 – page 16 line 27).

**Response to specific comments**

Our responses to specific comments are included below in blue.

**Anonymous reviewer #1**

1. How do the analysis presented here (and in Armitage, 2016) compare with other state estimates of the Arctic such as MIMOC, WOA, PHC or the NCEI climatology, or. eg., ocean data assimilative products? Do other analyses lead to essentially the same circulation, T, S, ice pattern changes over the last decade? Is it possible to estimate the error bars or uncertainty on your estimates somehow?

We have avoided making comparisons between our data and climatologies (e.g., MIMOC, WOA, PHC, NCEI) because in the Arctic, particularly in ice-covered regions and during the winter months, such comparisons will simply highlight insufficiencies of data included in the climatologies rather than provide any meaningful insight about our new data. Similarly, the lack of data going into ocean reanalysis products means that they will be more or less free running models in the ice-covered regions of the Arctic, and a comparison will simply serve to highlight this fact.

In Armitage et al. (2016) we estimated the uncertainty on the SSH data through a crossover analysis, and in the present manuscript we have quantified the uncertainty on the satellite derived currents through comparison with direct *in situ* measurements of the current speed under ice (Section 3.1, Figure 2, Table 1).

2. The paper is very descriptive and there is little discussion of underlying mechanisms. Can the authors delve a little deeper and do more than describe what is happening and reporting that 'so-and-so said this' etc. At the moment the paper is very value-neutral and not very deep. It would be good to have some viewpoint expressed.

In this paper, we are presenting and describing a brand new data set, and have quantified the variability in surface geostrophic currents over this time period, at seasonal to inter-annual timescales. We feel it is worth stressing that knowledge of geostrophic currents in the Arctic Ocean is extremely lacking, and we see the introduction of this new monthly dataset being the primary value of our study. We have attempted to put this new dataset into context within the current literature and have provided a discussion of how it fits in with the broader picture of Arctic Ocean environmental change. As you point out, this data is a good foundation for future analytical studies of the underlying mechanisms of change, for example pulling apart why we are really seeing the changes in ice drift speed, which has not yet been fully answered in our opinion. We are also unsure what exact viewpoint it is that you would like expressed.

3. There has been considerable effort by the community to measure time-series across key Straits - Fram, Davis etc. Why not compare the estimates derived here with those that are directly measured? Is it because one needs the Ekman transport too? – but that would be directly and readily given by the wind, mediated by ice. Instead we are presented with geostrophic transports through the mini-sections shown in Fig.4, sections that have not been instrumented and thus not directly measured. This is a rather serious shortcoming of the paper I believe.

We are aware of the mooring data, but there are a couple of reasons why we haven't performed this analysis. The moorings don't usually measure the surface currents, but are instrumented at 50m depth and below, so we would have to introduce a lot of assumptions about the velocity in the upper 50m (Ekman velocities and associated ice-ocean/air-ocean drag, turning angles, stratification) in order to derive surface geostrophic currents and make a like-for-like comparison. With the ADCP data available on the BGEP moorings we were at least able to look directly at the surface currents and not the current at a depth of 50m. Because of this we haven't gone to the considerable effort of synthesizing the data and attempting to derive surface currents. In Fram Strait for example, where the mooring data might be most useful, the moorings are deployed separately by AWI and the Norwegians, between deployments the mooring location and the instrument depths can change, and the data are then distributed as annual deployments so require synthesizing and editing to obtain a coherent time series.

4. The paper would benefit from a plot of timeseries of key metrics such as FWC, AOO, sea-ice extent etc etc, so that they can be compared with those presented by, e.g, Proshutinsky.

We do discuss the most pertinent of these climate indices, the AOO (page 14, line 22 – page 15 line 8), in order to draw comparison between our work and that of Proshutinsky. However, it is not clear to us how the paper would benefit from the addition of these time series, or how they would benefit the discussion presented in the text.

Fig.1 is key, could be a very useful figure, and yet not easy to read. The labels need to be more distinctive and the confluence of dark blue and dark grey is not easy to parse. Could this be redrawn with attention to labels, colors etc so that they can be more easily read.

Agreed. We have modified the colour scale and text on Figure 1 in order to improve the figure clarity.

Fig.7 also needs some attention. Many of the details cannot be discerned.

Agreed. We have modified the colour scale on Figure 7 to try to make the details more discernable.

**Anonymous reviewer #2**

Title: Should really be "...surface geostrophic circulation 2003-2014" Abstract: State what data is used to derive the velocity estimates.

Agreed, we have made these two changes (title page and page 2 line 4).

Figure 1: Poor choice of color scale, can this be made easier to read?

Agreed (anonymous reviewer #1 made the same point). We have modified the colour scale and text to try to improve this.

(page 3, line 4) Water and ice are not transported from the BG into the Chukchi Sea, it is the

other way around.

The work of Petty et al. (2016), "Sea ice circulation around the Beaufort Gyre: The changing role of wind forcing and the sea ice state,  J. Geophys. Res. Oceans, 121, doi:10.1002/2015JC010903", as well as the work of others, shows a net transport of ice out of the western Beaufort into the Chukchi (their Figure 6). Maps of ice drift, and the ocean currents presented in this paper show that the westward flow along the southern periphery of the Beaufort Gyre, along the northern coast of Alaska, show that the transport is westward, from the Beaufort Sea into the Chukchi Sea.

(4,1) Why can you do seasonal and interannual analysis while previous studies were limited to long term means?

This is explained early in the manuscript (page 4, lines 9-13) – "we calculate monthly geostrophic currents using monthly, satellite-derived estimates of DOT from the ice-covered and ice-free portions of the Arctic Ocean between 2003 and 2014". Other datasets were limited to long-term (annual or greater) averages or intermittent seasonal means. This is the crucial improvement made by the dataset presented in this paper.

 (4,28) The authors are correct that the ocean and ice coupling can go both ways, unfor-tunately they do not attempt to decompose which is driving the other in their analysis.

We agree, but making this decomposition is certainly not trivial (involving, amongst other things, assessing the role of changing atmosphere-ice-ocean drag, boundary layer physics) and is beyond the scope of the present manuscript other than to point out that the new data is highly relevant for this kind of future work.

(5,11) The Manley and Hunkins eddies are very small, O( 10 km), and thus not detectable in the present filtered satellite data. This is a very important distinction. The EKE estimated here is more likely dominated by the gyre instability, which has been linked to wind stress and freshwater content by Manucharyn and Spall (2015 GRL) and Manucharyan et al. (2016 JPO).

We have added further discussion of eddies based on the comments of another reviewer (page 15 line 26 – page 16 line 15). You are correct, and we have now made it clear in the discussion that we are observing larger-scale variability (~50km) with the satellite data (page 13 lines 6-9 & page 16 line 16 – page 16 line 27).

 (9,2) Looks more like the speed increased from 2008-2009.  (9,26) Is the direction difference partially explained by Ekman dynamics?

Thanks, we have corrected this in the revised manuscript (page 9 line 2). Yes, Ekman dynamics will play a role in the observed difference, as we discuss later in this section (page 9 line 13-16).

Figure 3: The vectors are very difficult to interpret (too small). I suggest contouring DOT together with the color plot of the velocity magnitude.

We have made the vectors larger and thicker in order to make the figure clearer. We anticipate this figure to be large in the printed manuscript which will also help.

(11,1) I find the "gate" approach ambiguous. How much of the changes are due to changes in magnitude versus changes in gyre location?

Whilst we accept that the choice of the gates is somewhat arbitrary we did locate them in order to try to characterize important aspects of the circulation, and in that aspect we believe the gate analysis provides a useful assessment of current variability and change around the Beaufort Sea, and through Fram Strait. In fact, from the changing circulation in Figure 3g-h and through the gates we realized that the location of the gyre had probably changed during the time period, hence why we added Figure 5, tracking the gyre centroid. Understanding how much of the changes are due to the changing location of the gyre will likely require a modelling study (e.g. fixing the Beaufort High location but allowing the strength to vary) which is beyond the scope of this paper.

(11,14) Does the BG extend over the Northwind Ridge when its centroid is located to the extreme NW? This is a little surprising to me.

In terms of the dome of DOT, yes, the gyre extends well over the Northwind ridge and onto the Chukchi Plateau in the latter period of this time series.

(12,1) Figs 3a, b look nearly the same to me. I would guess if one were to put error bars on these they would be the same.

Figures 3a & b are similar, however they do help demonstrate that the major circulation features are consistently stronger in winter than summer (see also Figure 6). It is hard to quantify the uncertainty on the monthly currents, however we have attempted to do so in Section 3.1 and Table 1. Taking a monthly RMS error of 2cm/s (the upper limit from Table 1), and considering that we are averaging 48 and 96 months for 'ice free' and 'ice covered' respectively, we should be sensitive to seasonal variations at a level of 0.2-0.3cm/s.

(12,13) In what way does the present study corroborate Bulczak et al?

In the revised text (page 12, lines 13-23) we have highlighted features in the EKE fields in the Nordic Seas that were first presented by Bulczak et al. (2015): The so-called Lofoten Basin eddy, the east-west differences in EKE in the Norwegian and Greenland Seas, and the high EKE running down east Greenland coincident with the sea ice edge and the shelf break.

(12) The numerous features of EKE that are mentioned are difficult to locate on the figure. Maybe some geographic indicators would help the reader. Again, the EKE found here is probably not the small Manley and Hunkins eddies generated from the shelf break. It would be interesting to try to connect the magnitude of this EKE to that predicted by Manucharyan's work.

As mentioned in response to your comment, we have added further discussion of eddies

based on the comments of another reviewer (page 15 line 26 – page 16 line 15) and we have now made it clear in the discussion that we are observing larger-scale variability (~50km) with the satellite data (page 13 lines 6-9 & page 16 line 16 – page 16 line 27).

(13,6) I do not see any trackiness in the figure.

Fair point. We believe this is more visible in the updated figure, in particular in the Barents Sea.

Discusson: The discussion is really a summary of other people findings and a recap of the present results connected by some speculation. I don't think it contributes much to the paper as is.

As we have said in our response to reviewer 1, we believe that framing this new dataset, and the variability/change seen during the time period, within the current literature is a useful exercise. A paper describing a new dataset set is necessarily somewhat descriptive.

(17,14) Fram Strait

Thanks, changed.

---

## Author Response (AR2)

**Response to the editors request for minor revision**

T. W. K. Armitage, S. Bacon, A. Ridout, A. Petty, S. Wolbach, M. Tsamados

*...I ask the authors to think again about point 3 of anonymous reviewer number 1. If the mooring data really cannot be used without introducing extra unconstrained assumptions, then the reasons why this data cannot be used should be given in the manuscript. Such reasons may provide insight into how such moorings could more usefully be deployed in the future or suggest an extra line of work to much such data useable.*

We have added a paragraph explaining why we have not utilized the available mooring data to Section 3.1: "Data Evaluation" (page 10, lines 3-20).